# Novel protein candidates for serodiagnosis of African animal trypanosomosis: Evaluation of the diagnostic potential of lysophospholipase and glycerol kinase from *Trypanosoma brucei*

**Magamba Tounkara**[1,2,3], **Alain Boulangé**[2,4,5], **Magali Thonnus**[1], **Frédéric Bringaud**[1],
**Adrien Marie Gaston Bélem**[6], **Zakaria Bengaly**[3], **Sophie Thévenon**[4,5],
**David Berthier**[4,5], **Loïc Rivière**[1] *

**1** Univ. Bordeaux, CNRS, Microbiologie Fondamentale et Pathogénicité, UMR 5234, Bordeaux, France,
**2** CIRAD, UMR INTERTRYP, Bobo-Dioulasso 01, Burkina Faso, **3** Centre International de Recherche-Développement sur l'Élevage en zone Subhumide (CIRDES), Bobo-Dioulasso 01, Burkina Faso, **4** CIRAD, UMR INTERTRYP, Montpellier, France, **5** INTERTRYP, Univ Montpellier, CIRAD, IRD, Montpellier, France, **6** Université Nazi Boni (UNB), Bobo-Dioulasso 01, Burkina Faso

* loic.riviere@u-bordeaux.fr

## Abstract

African trypanosomosis, a parasitic disease caused by protozoan parasites transmitted by tsetse flies, affects both humans and animals in sub-Saharan Africa. While the human form (HAT) is now limited to foci, the animal form (AAT) is widespread and affects the majority of sub-Saharan African countries, and constitutes a real obstacle to the development of animal breeding. The control of AAT is hampered by a lack of standardized and easy-to used diagnosis tools. This study aimed to evaluate the diagnostic potential of TbLysoPLA and TbGK proteins from *Trypanosoma brucei brucei* for AAT serodiagnosis in indirect ELISA using experimental and field sera, individually, in combination, and associated with the BiP C-terminal domain (C25) from *T. congolense*. These novel proteins were characterized *in silico*, and their sequence analysis showed strong identities with their orthologs in other trypanosomes (more than 60% for TbLysoPLA and more than 82% for TbGK). TbLysoPLA displays a low homology with cattle (<35%) and *Piroplasma* (<15%). However, TbGK shares more than 58% with cattle and between 45–55% with *Piroplasma*. We could identify seven predicted epitopes on TbLysoPLA sequence and 14 potential epitopes on TbGK. Both proteins were recombinantly expressed in *Escherichia coli*. Their diagnostic potential was evaluated by ELISA with sera from cattle experimentally infected with *T. congolense* and with *T.b. brucei*, sera from cattle naturally infected with *T. congolense*, *T. vivax* and *T.b. brucei*. Both proteins used separately had poor diagnostic performance. However, used together with the BiP protein, they showed 60% of sensitivity and between 87–96% of specificity, comparable to reference ELISA tests. In conclusion, we showed that the performance of the protein combinations is much better than the proteins tested individually for the diagnosis of AAT.

**Data Availability Statement:** All relevant data are within the manuscript and its Supporting Information files.

**Funding:** This work was funded by the Laboratoire d'Excellence (LabEx) "French Parasitology Alliance For Health Care" (ANR-11-LABX-0024-PARAFRAP, https://labex-parafrap.fr), programme PhD Pays du sud (including salary of Magamba Tounkara). Experiment costs were also supported by Université de Bordeaux (https://www.u-bordeaux.fr), CNRS (https://www.cnrs.fr), CIRAD (https://www.cirad.fr) and CIRDES (Centre International de Recherche-Développement sur l'Élevage en zone Subhumide, https://www.cirdes.org). The funders had no role in study design, data collection and analysis, decision to publish or preparation of the manuscript.

**Competing interests:** The authors have declared that no competing interests exist.

## Author summary

African animal trypanosomiasis (AAT) is an endemic disease in sub-Saharan Africa that hinders the development of livestock production on the continent. The control of the disease is based on chemotherapy, vector control and diagnosis. Misuse, as well as the continuous/regular use of a limited number of anti-trypanosomal drugs, is responsible for the appearance of increasingly drug-resistant strains of trypanosomes. In terms of serological diagnosis, the most efficient test at present suffers from a lack of reagent standardization. Unfortunately, even the most promising candidates fail due to low sensitivity in primately or chronically infected animals. Based on this observation it seems obvious that diagnosis must be revisited. In this study we evaluated the diagnostic potential of two *Trypanosoma brucei* proteins, TbLysoPLA and TbGK, in indirect ELISA for antibody detection. To provide a proof of concept that the judicious association of immunoreactive proteins could improve the sensitivity and specificity of tests based on recombinant antigens, we used these molecules alone and then in combination, associated or not with the BiP protein of *T. congolense*. The evaluation in serological diagnosis showed that the two proteins used separately had a poor performance. However, when used together with the BiP protein, they showed a sensitivity of 60% and a specificity between 87 and 96%, comparable to the reference tests. It shows for the first time that the performance of protein combinations is much better than that of the proteins tested individually for the diagnosis of AAT.

## Introduction

African animal trypanosomoses (AAT) or nagana are hemoparasitic infections caused mainly by *Trypanosoma congolense*, *T. vivax* and *T. brucei brucei* that are transmitted mostly by tsetse flies. Nagana is a devastating livestock disease that affects 38 African countries in the sub-Saharan region. Nearly 55 million cattle are at risk over an area of 10 million km$^2$ where the tsetse fly is endemic [1]. The clinical manifestations of the disease are diverse and non-pathognomonic. Observed symptoms include weight loss, sterility, abortion, and in some acute cases the rapid death of the animal. The major pathological feature is anemia [2]. This disease leads to a general deterioration of the animal conditions that diminishes its physical performance and ability to produce milk and meat, thus causing both economic and nutritional difficulties for the growing African population. According to the FAO (2020) [3], nearly three million cattle die from AAT, and the overall economic losses are estimated at US$4.75 billion per year. Due to the ability of trypanosomes to regularly change their glycoprotein coat, conventional vaccines have not yet been developed against AAT [4], although it should be noted that a very recent study has shown the efficacy of an anti-*T.vivax* vaccine in mice [5].Therefore, the control of the disease is based on three pillars, vector control, chemotherapy and diagnosis. The continuous and often improper use of the same trypanocides since the 1960s, estimated by the FAO at 35 million doses per year in sub-Saharan Africa [3], is responsible for the emergence of drug resistances [6]. In this context, the detection of trypanosome infection becomes an essential step in chemotherapy. It is based on either of two principles: the detection of the parasite itself or its constituents, DNA, RNA or proteins, or the detection of the host immune response directed against the parasite, chiefly antibodies. Depending on the target, different methods can be used. Parasitological diagnosis based on microscopy is comparatively easy to perform but lacks sensitivity. Molecular diagnosis (PCR and LAMP) is both very sensitive and specific, but requires relatively sophisticated facilities [7]. Immunodiagnosis is the only one adapted for use in field conditions, particularly under RDT (Rapid Diagnostic Test) format. However,

there is as yet no antigen capture test available, that would allow detecting active infections. The antibody detection tests in ELISA format currently in use are limited by the use of whole trypanosome lysates as antigen [8], stumbling upon the problem of antigen standardization. The few tests that were developed making use of a unique recombinant protein as antigen to bypass standardization issues consistently lacked sensitivity. There is therefore a need for novel antigen candidates to increase the specificity and sensitivity of antibody-detection tests based upon recombinant proteins. Here we evaluated two proteins, the *T.b. brucei* lysophospholipase (TbLysoPLA) and the glycerol kinase (TbGK) for their potential in antibody-ELISA (Ab-ELISA).

Phospholipases are suspected to be involved (with other enzymes) in host erythrocytes lysis during trypanosomes infection [9–11]. They are involved in lipids catabolism and may constitute virulence and pathogenicity factors by affecting cell membrane lipids [9,10]. Trypanosomes possess several phospholipases. It has been shown that *T.b. brucei* lysophospholipase (TbLysoPLA) is a secreted/excreted protein, and that it induces the production of specific antibodies in mice infected with this parasite (Monic *et al.*, manuscript in revision). Because of this immunogenicity, TbLysoPLA was deemed suitable as a candidate for serological diagnosis, hence our choice for its evaluation in AAT diagnosis.

It has been shown that GK of *T.b. brucei* is expressed in large excess in BSF. When maintaining a high level of glycerol metabolic flux, the GK become essential for BSF [12]. The GK is glycosomal, it represents with other known glycolytic enzymes up to 4% of the trypanosome proteome [13] and it plays very critical functions in the cell. Several glycolytic enzymes have been targeted as antigens for the detection of various parasites. For instance, the frucose-1,6-biphosphate aldolase (FBPA) has been used for malaria [14] and schistosomiasis [15] immunodiagnostic. More recently, aldolase [16] and pyruvate kinase [17] have been targeted for the detection of *T. congolense* using specific antibodies from camelids. Glycerol kinase from *T.b. brucei* has been proposed as a therapeutic target [18]. In addition, preliminary studies in the laboratory had shown that specific antibodies directed against this protein are detected in *T.b. brucei* experimentally infected mice sera. This immunogenic property coupled with its intrinsic abundance guided our choice to evaluate TbGK as a diagnostic target.

The two proteins were tested individually, in combination with each other, and associated with a known immunodominant *T. congolense* antigen, the C-terminal part of the Immunoglobulin Binding Protein (BiP), called C25 [19]. These proteins selected for their immunogenicity or abundance, were characterized *in silico*, recombinantly expressed and evaluated for their diagnostic potential. Proteins mixtures showed a better diagnostic accuracy when compared to individual proteins in indirect ELISA. This result constitutes the first report of the superior performance of protein combinations in AAT diagnosis. It provides the opportunity to explore innovative approaches through the association of immunogenic peptides from immunoreactive proteins.

## Materials and methods

### Comparative sequence analysis and immuno-informatics

For each protein candidate, sequences were retrieved from the TriTrypDB database (https://tritrypdb.org). Orthologous sequences in other trypanosomes were obtained by performing a BLASTP search in the same database. Sequences in cattle, *Piroplasma spp* and *Anaplasma* were retrieved by a BLASTP search on Uniprot (https://www.uniprot.org/). The sequences were then aligned in pairs using Geneious v4.8 software (demo version) to determine their level of homology. Protein antigenicity was predicted using the Bepipred Linear Epitope Prediction 2.0 algorithm from the IEDB database (http://tools.iedb.org/bcell/).

## Total parasites lysates, TcoCB1 and C25 antigens for ELISA

The total lysates (of *T.b. brucei*, *T. congolense* and *T. vivax*) used as reference antigens in ELISA tests as recommended by the OIE (https://www.oie.int/en/international-standard-setting/terrestrial-manual/access-online/) were provided by CIRDES. Antigens were prepared as follows. Briefly, parasites were grown *in vivo* in rats, the blood collected at peak parasitemia, and parasites separated from the blood components by anion exchange chromatography on DEAE cellulose [20]. The purified trypanosomes were then lysed by three freeze/thaw cycles and sonicated in phosphate-saline-glucose buffer. Soluble antigens for ELISA were separated from the particulate elements by centrifugation. The recombinant cathepsin B-Like protease 1 (TcoCB1) antigen had been previously characterized and tested for its diagnostic potential [21]. It was recently included in a RDT device [22]. The purified TcoCB1 was provided by the UMR-5234-MFP laboratory (Bordeaux, France). We used this antigen as a control in our study. The C25 is a truncation of the Immunoglobulin Binding Protein (BiP) of *T. congolense* [19]. It had been cloned previously in the pMal-CRI plasmid in frame with the maltose binding protein (MBP)-tag [19]. The recombinant plasmid was used to transform *Escherichia coli* BL21 Star (DE3) strain (Invitrogen, Carlsbad, CA, United States). C25 was expressed and purified on amylose resin according to the manufacturer's instructions (Amylose Resin, NEB, UK).

## Cloning of TbLysoPLA and TbGK genes

The TbLysoPLA gene had been previously cloned in *T.b. brucei* phospholipases study context (Monic & al, manuscript in revision). Briefly, the TbLysoPLA open reading frame (ORF) (Tb427.08.6390) was amplified by PCR from *T.b. brucei* 427 genomic DNA using specific primers: aS (TbLysoPLA-pGEX-Fw): 5'-AGCTCGAGATGTTTGGCACGCCGTTGAGAAT-3'; aAS (TbLysoPLA-pGEX-Rv): 5'-AGGCGGCCGCTTACGATTTCGATGAAGGTCCGG-3'. The 861 bp product was cloned using the restriction enzymes Xho1 and Not1 (sites included in the primer sequences) into the plasmid pGEX-4T-1 (GE Healthcare 28-9545-49) in frame with the GST (Glutathione- S-Transferase) after the thrombin cleavage site. The TbGK (Tb927.9.12630) ORF had been previously cloned in the pET151 plasmid at the University of Tokyo [18], and was provided by the UMR-5234-MFP laboratory. The recombinant vectors have been used to transform *Escherichia coli* BL21 Star (DE3) (Invitrogen, Carlsbad, CA, United States) and JM109 (DE3) (Promega, Fitchburg, WI, United States) strains for TbLysoPLA and TbGK expression respectively.

## Expression and purification of TbLysoPLA and TbGK

The bacteria transformed with the recombinant vectors were cultured at 37˚C up to an Optical Density (OD) at 600 nm between 0.6 and 0.8. Protein expression was then induced by the addition of isopropylthio-β-D-galactoside (IPTG) to a final concentration of 1 mM. Protein production was continued for 3h at 37˚C under 200 rpm agitation. Bacteria were then harvested by centrifugation at 4000 x g for 10 min, and the pellets stored at -20˚C. For the TbLysoPLA protein purification the protocol was the same as described elsewhere (Monic & al, manuscript in revision). Pellets were suspended in PBS containing 100 μg/L lysozyme, the suspension incubated at 30˚C for 30 min under 200 rpm agitation, and the bacteria then lysed on ice in a sonicator (four cycles of 20 pulses for 30 s). The lysate was clarified by centrifugation at 10,000 x g for 30 min at 4˚C. The supernatant was loaded onto a glutathione Sepharose 4B resin column (GE Healthcare, Chicago, IL, United States) previously equilibrated with PBS that was placed on a rotator at 6 rpm overnight at room temperature (RT). The column was then washed with ten times volume of PBS using gravity only. Five units of bovine thrombin protease (Merck, Darmstadt, Germany) were added to the column, which was put back on the

rotator overnight at 30 rpm at RT. The slurry was left to sediment, and fractions containing the cleaved TbLysoPLA protein collected by gravity and stored at 4°C with 1x protease inhibitor cocktail (Roche, Basel, Switzerland). With regards to the TbGK protein, it was purified by affinity chromatography on the HisPur Ni-NTA nickel resin (Thermo Fisher Scientific, Waltham, MA, USA) according to the manufacturer's instructions. Protein concentrations were determined using the Pierce 660nm Protein Assay Kit (Thermo Fisher Scientific). Purified recombinant proteins were lyophilized to be transferred to CIRDES (Burkina Faso). All steps were verified by polyacrylamide gel electrophoresis (SDS-PAGE).

## SDS-PAGE

Polyacrylamide gel electrophoresis was done according to the following protocol. The samples were prepared by mixing 30 μL of each sample with 10 μL of 4x loading buffer (Tris-HCl 250 mM, SDS 8%, Glycerol 40%, 5 mg of bromophenol blue and β-mercaptoethanol 1.43 M). The mixture was boiled at 100°C for 10 min. Then 15 μL of each sample were loaded on Mini-PROTEAN TGX 4–20% precast gels (Bio-Rad, Hercules, CA, USA). The proteins were separated at 200 V for 30–45 min in Tris-Glycine-SDS buffer (TGS) 1X. After migration, the gel was washed with distilled water and stained with InstantBlue (Abcam, Cambridge, UK). The gel was then imaged with ImageQuant LAS 4000 (Cytiva, Washington, DC, USA).

## Western blotting

Total trypanosome extracts of *T.b. brucei* Tb427 strain and *T. congolense* IL3000 strain were prepared for western blotting by sonicating 5 x10^6 cells twice 30 s in 10 μL SDS 2% with protease inhibitor cocktail 1x. The suspension was mixed with 2x loading buffer and boiled for 10 min. TbLysoPLA *T.b. brucei* gene knockout strain (KOTbLysoPLA) extract and total *T.b. brucei* extract with RNAi silenced TbGK expression provided by the UMR-5234-MFP laboratory were also prepared. After separation on a 4–20% SDS-PAGE gel (Mini-PROTEAN TGX precast gels, Bio-Rad), the proteins were transferred to a polyvinylidene fluoride membrane (Trans-Blot Turbo Midi-size PVDF membrane, Bio-Rad) with the Trans-Blot Turbo Transfer System (Bio-Rad). The membrane was blocked for 1h at RT with PBS containing 0.1% Tween-20 (PBST) and 5% skimmed milk. The blocking buffer was removed and the membrane incubated overnight at 4°C with the primary antibody, either a polyclonal anti-TbLysoPLA or anti-TbGK produced by immunization of rabbits with recombinant proteins, diluted respectively at 1:1000 and 1:5000 in blocking buffer. The rabbit anti-PFR (anti-Paraflagellar Rod protein, diluted at 1:10 000) was used as loading control. The membrane was washed three times 10 min with 5 mL PBST, and incubated 45 min at RT with a secondary peroxidase-conjugated antibody diluted at 1:10 000 in blocking buffer. The membrane was washed as before and the reaction revealed by chemiluminescence using Pierce ECL Western Blotting Substrate (Thermo Fisher Scientific). Images were acquired with the ImageQuant LAS 4000.

## Origin of the sera used in the study

Two collections of sera were provided by CIRDES. One came from an experimental infection conducted in 2014 by GALVmed (Global Alliance for Livestock Veterinary Medicine) to study the efficacy of novel trypanocides. Animals had been experimentally infected with strains of *T. congolense* and *T. vivax* isolated in different West African countries. We tested 100 serum samples taken before infection and 100 taken post-infection but before treatment with trypanocides, on *T. congolense* indirect ELISA, of which 37 *T. congolense*-infected cattle samples and 31 non-infected cattle samples were used. The second CIRDES collection were field sera collected during the AAT-2019 mission, aimed at surveying the epidemiology of AAT in some

localities in south-west Burkina Faso (manuscript in preparation). ELISA and PCR data were available and served as a basis to select 37 positive sera from cattle with single and mixed infection with *T.b. brucei*, *T. congolense* and/or *T. vivax*, and 31 negative sera. In addition, CIRDES provided four sera from non-infected cattle, ten from *T.b. brucei*-infected cattle.

## Antibody detection by ELISA: Ab-ELISA

The ELISA antibody detection technique was used to evaluate the antigenicity of recombinant proteins. The protocol was optimized for each antigen to determine the optimal coating buffer, antigen coating concentration, and secondary antibody (conjugate) concentration. Nunc Immuno F96 Polysorb 96-well plates (Thermo Fisher Scientific) were coated with 0.5 μg of antigen per well in 100 mM bicarbonate buffer pH 9.6 at 4°C overnight (100 μL/well). The day after, each well was blocked with PBS-Tween-20 0.1% (PBST) with 5% skimmed milk (Régilait, Saint-Martin-Belle-Roche, France) for 1h at 37°C (150 μL/well). The blocking buffer was removed, each serum samples diluted 1:100 in blocking buffer added in duplicate wells (100 μL/well) and incubated for 1h at 37°C. Plates were then washed 3 times 5 min with PBST, and incubated for 1h at 37°C with 100 μL/well of a secondary antibody diluted in blocking buffer. Horseradish peroxidase-conjugated sheep anti-bovine IgG (ref. AAI23P, Bio-Rad) diluted 1:10 000 was used. After a washing step as above, 100 μL/well of Enhanced K-Blue Substrate (Neogen, Lansing, MI, USA) was added and the plates were incubated for at least 15 min in the dark. The OD was measured at 620 nm in an ELISA reader.

## Statistics and results analysis

Image J, Excel version 2011, GraphPad Prism version 5, and R software were used to analyze and process the results. For ELISA, the duplicate OD values of each sample were averaged, the horseradish peroxidase-conjugated antibody OD value subtracted from it, and expressed as relative percent of positivity RPP = (sample OD/positive control OD) x 100 to normalize antigen reactivity. The non-parametric Mann-Whitney test was used to assess the significance of the difference of RPP between infected and non-infected cattle sera for the experimental sera, and positive versus negative sera samples for field sera. In addition, Receiver Operating Characteristic (ROC) analysis was conducted using GraphPad Prism software to determine the diagnostic performance parameters for each antigen at the optimal cut-off value. These parameters include the area under the curve (AUC), the sensitivity (Se), the specificity (Sp) and their 95% confidence interval (CI), and finally the likelihood ratio (LR). The cut-off value for each serum panel was determined using the method described by Johnson (2004) [23] based on the likelihood ratio closer to the maximum vertical distance method [24]. Samples with RPP values above the cut-off were categorized positive for the test and samples with RPP values below the cut-off were categorized negative.

## Results

### Protein sequences analysis and prediction of antigenicity

To assess a possible cross-reaction with other blood pathogens such as *Piroplasma* parasites or *Anaplasma spp*. and predict the antigenicity of both TbLysoPLA and TbGK, sequence alignments and B-cell epitope predictions were conducted. The sequences analysis also allowed to evaluate the likelihood of cross reactivity between trypanosome species and with *Leishmania*. Each protein was compared to its orthologs (in *T. congolense* and *T. vivax*) and the relative percentage of homology was determined by pairwise alignment of the amino acid sequences. The results show that phospholipase TbLysoPLA is conserved in trypanosomes group with more

**Table 1. Percentage of homology of TbLysoPLA and TbGK and their orthologs.**

| Protein | T.b. brucei | T. congolense | T. vivax | Leishmania infantum | Bos taurus | Babesia bovis | Theileria parva | Species |
|---------|-------------|---------------|----------|---------------------|------------|---------------|-----------------|---------|
| TbLysoPLA a | - | 71.9 | 62.4 | 53.0 | 32.1 | 5.30 | 10.4 | T. b brucei |
| | | - | 61.8 | 53.8 | 34.0 | 12.9 | 6.80 | T. congolense |
| | | | - | 51.9 | 34.2 | 6.30 | 20.0 | T. vivax |
| TbGK b | - | 88.3 | 82.0 | 60.4 | 58.3 | 45.3 | 55.8 | T. b brucei |
| | | - | 82.0 | 59.6 | 57.8 | 45.6 | 38.6 | T. congolense |
| | | | - | 57.3 | 57.3 | 46.2 | 37.6 | T. vivax |

a T. b brucei lysophospholipase. b T. b brucei glycerol kinase

than 61% homology, displaying the most identity with its ortholog in *T. congolense* with 71.9% homology. The three orthologs share about 53% homology with *Leishmania infantum* but show a weak identity with those in cattle (<35%) and *Piroplasma* parasites (identity <15%) (Table 1). TbGK is much more conserved in trypanosomes (homology ≥82%). The trypanosomes glycerol kinase share a high identity (about 60% homology) with bovine and *L. infantum* orthologs but also with those of *Babesia bovis* (≥45%) and *Theileria parva* (~55% with *T. b. brucei*, ≤39% with *T. congolense* and *T. vivax*) (Table 1). The prediction of epitopes for TbLysoPLA and TbGK reveals presence of seven linear epitopes on the TbLysoPLA sequence spread over the entire sequence and ranging in size from 6 to 27 amino acids (aa) and scoring between 0.529 to 0.619 (Table 2). Five of those epitopes are relatively conserved in *T. congolense* ortholog protein and unique to trypanosome proteins when compared to their orthologs in *Piroplasma*, *Leishmania* and Bovine (S3 and S4 Figs). The TbGK protein shows 14 linear epitopes of varying size from 5 to 48 aa with scores ranging from 0.469 to 0.537 (Table 2). About 13 predicted epitopes in TbGK sequence are conserved in *T. congolense* ortholog and 12 in *T. vivax* (S5 Fig). In contrast to TbLysoPLA, the predicted epitope 10 of TbGK is conserved in *Leishmania* ortholog protein (S6 Fig).

## Specific anti-TbLysoPLA and anti-TbGK antibodies react respectively with TbLysoPLA and TbGK orthologs in *T. congolense*

Comparative analysis of TbLysoPLA and TbGK sequences showed a relatively good conservation within the *Trypanosoma* genus. It may thus be possible to use these proteins to detect trypanosomal infections across the species spectrum. To confirm this hypothesis, we used specific anti-TbLysoPLA and anti-TbGK antibodies on western blot of total extract of *T.b. brucei* and *T. congolense* wild-type (WT), on extract of TbLysoPLA *T.b. brucei* gene knockout strain (KOTbLysoPLA), and on total *T.b. brucei* extract with RNAi silenced TbGK expression (Fig 1A and 1B). Anti-TbLysoPLA antibody specifically detected TbLysoPLA protein in WT *T.b. brucei*, but also its ortholog in *T. congolense* at approximately 30 kDa, with comparable intensity when normalized with the loading control PFR (Paraflagellar Rod) protein. No specific band was observed in the KOTbLysoPLA strain extract (Fig 1A). Using anti-TbGK antibodies, a band at the expected size slightly above 55 kDa was observed in both extracts of *T.b. brucei* and *T. congolense* WT parasites. It was absent in the *T.b. brucei* TbGK RNAi extract (Fig 1B).

## Expression and purification of recombinant proteins

TbLysoPLA and TbGK fused to GST and the "hexa-histidine tag, respectively, were purified by affinity chromatography (Fig 2). The TbLysoPLA protein cleaved off the glutathione column showed a yield of 1 mg for 5 mL of initial bacterial culture. The purity was very high-quality as

**Table 2. Antigenic peptides of TbLysoPLA and TbGK predicted using BepiPred-2.0 prediction.**

| No | Start | End | Peptide | Length | Score |
|----|-------|-----|---------|--------|-------|
| | | | *T.b. brucei* **lysophospholipase a** | | |
| 1 | 7 | 12 | ENLTAN | 6 | 0.529 |
| 2 | 29 | 55 | GCIEQKDLQEKLEKTRETELITHGLKY | 27 | 0.583 |
| 3 | 60 | 66 | QIGNRQS | 7 | 0.546 |
| 4 | 107 | 117 | MQPVGINGGAV | 11 | 0.576 |
| 5 | 122 | 139 | YDIRNVSSGNGVTEDAEA | 18 | 0.619 |
| 6 | 202 | 211 | EKILPQLCNK | 10 | 0.556 |
| 7 | 250 | 260 | SYPMEHSSHPK | 11 | 0.537 |
| | | | *T.b. brucei* **glycerol kinase** [b] | | |
| 1 | 22 | 49 | RQRPVSVHQVPHTQHTPHPGWLEHDPME | 28 | 0.515 |
| 2 | 63 | 74 | AKLRQKDASFRK | 12 | 0.490 |
| 3 | 93 | 97 | VTKEP | 5 | 0.469 |
| 4 | 107 | 131 | LRTYDITKKVTAELGGGDSMFASKI | 25 | 0.503 |
| 5 | 153 | 165 | PAVADACRRGTLC | 13 | 0.473 |
| 6 | 197 | 244 | DLRTRKWSPELCEKLKIPMETLPEIRSNSELFGYVETDECGVAAALNE | 48 | 0.517 |
| 7 | 285 | 294 | GEEARFSKHG | 10 | 0.537 |
| 8 | 303 | 308 | VGRDGP | 6 | 0.492 |
| 9 | 328 | 350 | RRNMNLFSHITECEKLARSVPGT | 23 | 0.528 |
| 10 | 365 | 370 | PYWDPS | 6 | 0.481 |
| 11 | 377 | 383 | GMTLKTT | 7 | 0.493 |
| 12 | 408 | 414 | RDAGLNL | 7 | 0.491 |
| 13 | 441 | 447 | ILVPSMH | 7 | 0.482 |
| 14 | 465 | 509 | WTSLEEVKAVSRRENSWKTVSPSGSAMEREAMIAEWREALKRTKW | 45 | 0.533 |

a Threshold at 0.52. b Threshold at 0.45

shown by SDS-PAGE (Fig 2A). TbGK was eluted from the Ni-NTA column by competition with imidazole at high concentration (Fig 2B). The production yield of TbGK was 1–2 mg for 5 mL of initial bacterial culture. The last collected fractions of TbGK were also of quality purity as shown in Fig 2B. Finally, the concentration of purified proteins obtained was adequate for use in ELISA antibody detecting test (Ab-ELISA).

## Assessment of the diagnostic potential of TbLysoPLA and TbGK taken individually

The reactivity of TbLysoPLA and TbGK were tested with different cattle sera from: 1/ *T. congolense* experimentally infected cattle (TcEIC); 2/ *T. congolense*, *T.b. brucei* and *T. vivax* naturally infected cattle from the field (FS); 3/ *T.b. brucei* experimentally infected cattle (TbbEIC). A total of 74 positive sera (37 TcEIC and 37 FS), 62 negative sera (31 naive TcEIC sera and 31 FS sera) and 14 TbbEIC samples (10 positives and 4 naives) all randomized, were tested by Ab-ELISA. *T.b. brucei* whole trypanosome lysate (TbWTL), *T. congolense* whole trypanosome lysate (TcWTL), and TcoCB1 antigen [21], were all used as reference antigens. The optimal cut-off value allowing to differentiate positive from negative samples was fixed for each protein and control antigen as detailed above.

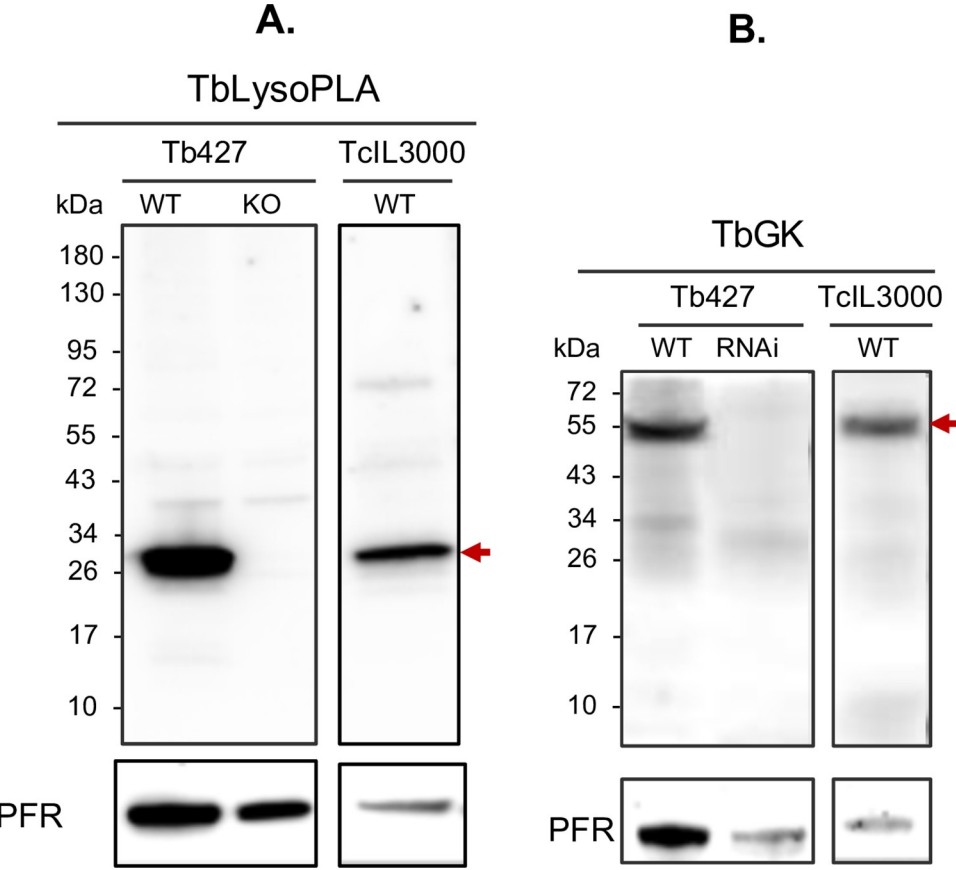

**Fig 1. Cross-detection of TbLysoPLA and TbGK orthologs on T. congolense total extract by western blot.** Western blot analysis of TbLysoPLA (A.) and TbGK (B.) in wild type (WT) parasites (Tb427 and TcIL3000), in ΔTbLysoPLA KO (knockout of TbLysoPLA gene) and ΔTbGK RNAi (RNA interference of TbGK mRNA in Tb427). The PFR (Paraflagellar Rod) protein was used as loading control.

- *Experimentally* T. congolense-*infected bovine sera reactivity to TbLysoPLA and TbGK*

ELISA results normalized and presented as RPP (Relative Percent of Positivity) from TcEIC samples show that TbLysoPLA protein is specifically recognized by positive (infected) sera with a RPP median value of twelve, which is higher than the RPP median value of negative (non-infected) sera (4) (Table 3). Statistical analysis comparing the two median values showed very significant difference (p = 0.0007) demonstrating that TbLysoPLA discriminates between infected and non-infected cattle groups (Table 3, Fig 3A). For the TbGK protein, there was no significant difference (p = 0.08) in RPP medians values between positive and negative sera (Fig 3A and Table 3) compared to TcoCB1, TbWTL, and TcWTL, which showed highly significant differences (p<0.0001) between sera from infected and non-infected cattle groups (Fig 3B).

- *Naturally* T.b. brucei, T. congolense and T. vivax *infected bovine sera reactivity*

TbLysoPLA and TbGK showed slightly different reactivity between positive and negative samples when using field sera (FS). This difference was statistically significant by Mann-Whitney test comparing the RPP medians of positive to negative sera. The p values were 0.03 for both TbLysoPLA and TbGK (Table 3), lower than 0.05, but the discrimination between groups of positive and negative sera was not clear-cut (Fig 3C). The RPP medians of positive samples and negative samples were lower (Table 3) than the respective cut-off values of 28 and 50 for

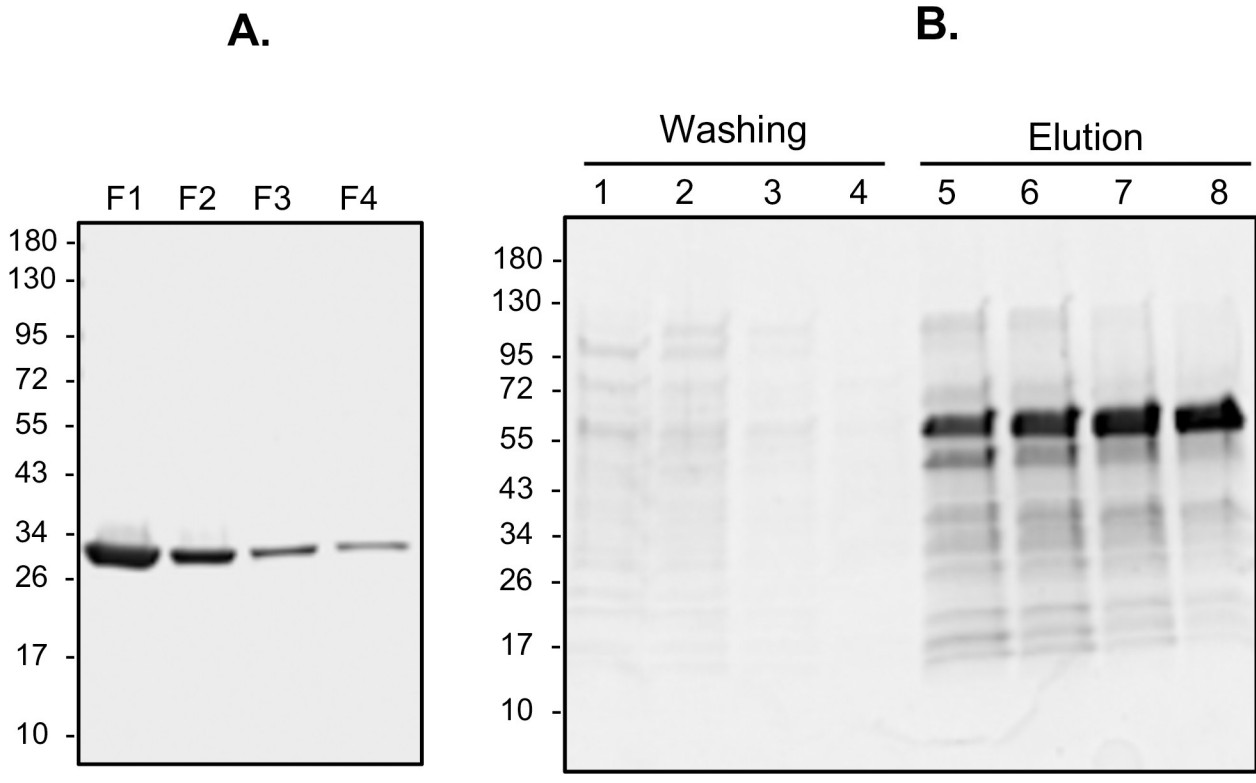

**Fig 2. TbLysoPLA and TbGK proteins purification.** Purification of TbLysoPLA (A.) by affinity chromatography on GE glutathione sepharose 4B. Lanes F1-F4 correspond to the collected fraction after GST release by thrombin cleavage. Nickel affinity purification of TbGK-6xHis fusion protein (B.) on HisPur Ni-NTA resin. Lanes 1–4: washes; lanes 5–8: imidazole elution fractions.

TbLysoPLA and TbGK in Table 5, respectively. As shown in Table 3, TbLysoPLA had RPP medians of 8 for negative samples and 13 for positive samples. The RPP medians for TbGK were in the same range, at 9 for negative samples and 11 for positive ones. Although these sera are well discriminated on total antigens (TbWTL and TcWTL) with a strong reactivity of the positive samples, TcoCB1 shows the same pattern of response as TbLysoPLA and TbGK (Fig 3D).

- *Experimentally* T.b. brucei-*infected bovine sera reactivity to TbLysoPLA and TbGK*

Statistical power of the tests was lower when using TbbEIC samples than using TcEIC and FS due to the smaller number available. Even so, the reactivity of *T.b. brucei* positive bovine sera were specific and significantly different from negative samples for both TbGK ($p = 0.004$) and TbLysoPLA (slight weak, $p = 0.03$) proteins (Table 3). The RPP medians are shown in S2 Fig.

## Diagnostic potential of TbLysoPLA and TbGK as a mixture, and combined with C25

Several studies have shown that the main limitation of single protein based assays is often lack of sensitivity [19,22,25,26]. As expected, this pattern of response is equally present for the two tested proteins. And as previously reported, both TcoCB1 [22] and C25 [19] displayed lower sensitivities with both field sera (Figs 3D and S1F and TcoCB1) and experimental sera (S1A– S1C Fig and C25_SE). In field sera, C25 showed better performance to discriminate infected

**Table 3. Statistical analysis of infected vs. non-infected samples comparison for single protein and mixture testing.**

| Evaluated antigens | TbGK | | TbLysoPLA | | TbLysoPLA_TbGK | | TbLysoPLA_TbGK_C25 | |
|---|---|---|---|---|---|---|---|---|
| | *T. congolense* experimentally infected cattle sera | | | | | | | |
| Samples status | Non-infected | Infected | Non-infected | Infected | Non-infected | Infected | Non-infected | Infected |
| RPP median (n) | 13 (31) | 23 (37) | 4 (31) | 12 (37) | 17 (31) | 34 (37) | 10 (31) | 21 (37) |
| p-value | 0.08 ns | | 0.0007 *** | | <0.0001 **** | | <0.0001 **** | |
| | *T. congolense, T.b. brucei* and *T. vivax* naturally field infected cattle sera | | | | | | | |
| RPP median (n) | 9 (31) | 11 (37) | 8 (31) | 13 (37) | 17 (31) | 39 (37) | 4 (31) | 23(37) |
| p-value | 0.03 * | | 0.03 * | | 0.0007 *** | | <0.0001 **** | |
| | *T. congolense, T.b. brucei* and *T. vivax* naturally field infected cattle sera | | | | | | | |
| RPP median (n) | 74.5 (4) | 106 (10) | 18 (4) | 31 (10) | 2 (4) | 3 (10) | 15.5 (4) | 42 (10) |
| p-value | 0.004 ** | | 0.03 * | | 0.52 ns | | 0.0020 ** | |

RPP (Relative percentage of positivity), (*n*): number of tested samples

p-value calculated by Mann-Whitney non-parametric test.

ns (no significant)

*(weak significant difference)

**(significant difference)

***(very significant difference)

****(highly significant difference).

and non-infected cattle groups (S1B–S1D Fig). Thus, we sought to assess whether we could potentiate the sensitivity of TbLysoPLA and TbGK based tests through combining these two proteins, with each other in the first place, and then with C25. Fig 4 shows ELISA results of the two tested mixtures. Equimolar mix of TbLysoPLA and TbGK discriminated significantly the sera of infected cattle from those of non-infected cattle groups (Fig 4A–4C and Table 3). With TcEIC sera, the p value was p<0.0001 for both TbLysoPLA_TbGK (a) and TbLyso-PLA_TbGK_C25 (b) mixtures when comparing positive and negative sera (RPP medians value shown in Table 3). The RPP medians of the positive sera, 34 for a and 21 for b, were higher or closer to the respective cut-off values, while RPP medians of negative sera, 17 for a and 10 for b, were smaller (Fig 4A). Similar results were observed with FS sera (Field sera, Fig 4C and Table 3). With the TbbEIC sera, TbLysoPLA_TbGK mixture did not show any difference between positive and negative samples (p = 0.52) while TbLysoPLA_TbGK_C25 combination could discriminate sera of *T.b. brucei* infected cattle groups from non-infected ones (p = 0.002, Table 3 and S2 Fig). As mentioned above however, statistical power was lower because of the fewer samples tested.

## Comparison of global diagnostic performance of individual protein testing *vs*. mixtures based testing

By plotting the RPP values and comparing positive to negative samples, the sera reactivity to protein mixtures is higher than that of proteins individually tested. To compare the global diagnostic performance of the different tests, we did a ROC analysis that allowed to determine for each ELISA test the sensitivity (Se), specificity (Sp), likelihood ratio (LR) and the area under the curve (AUC) (Tables 4 and 5). The LR value presented for each protein corresponds to the highest value, leading to the determination of the optimal cut-off value that efficiently discriminate positive from negative sera (Tables 4 and 5). The ROC curve expressing the true

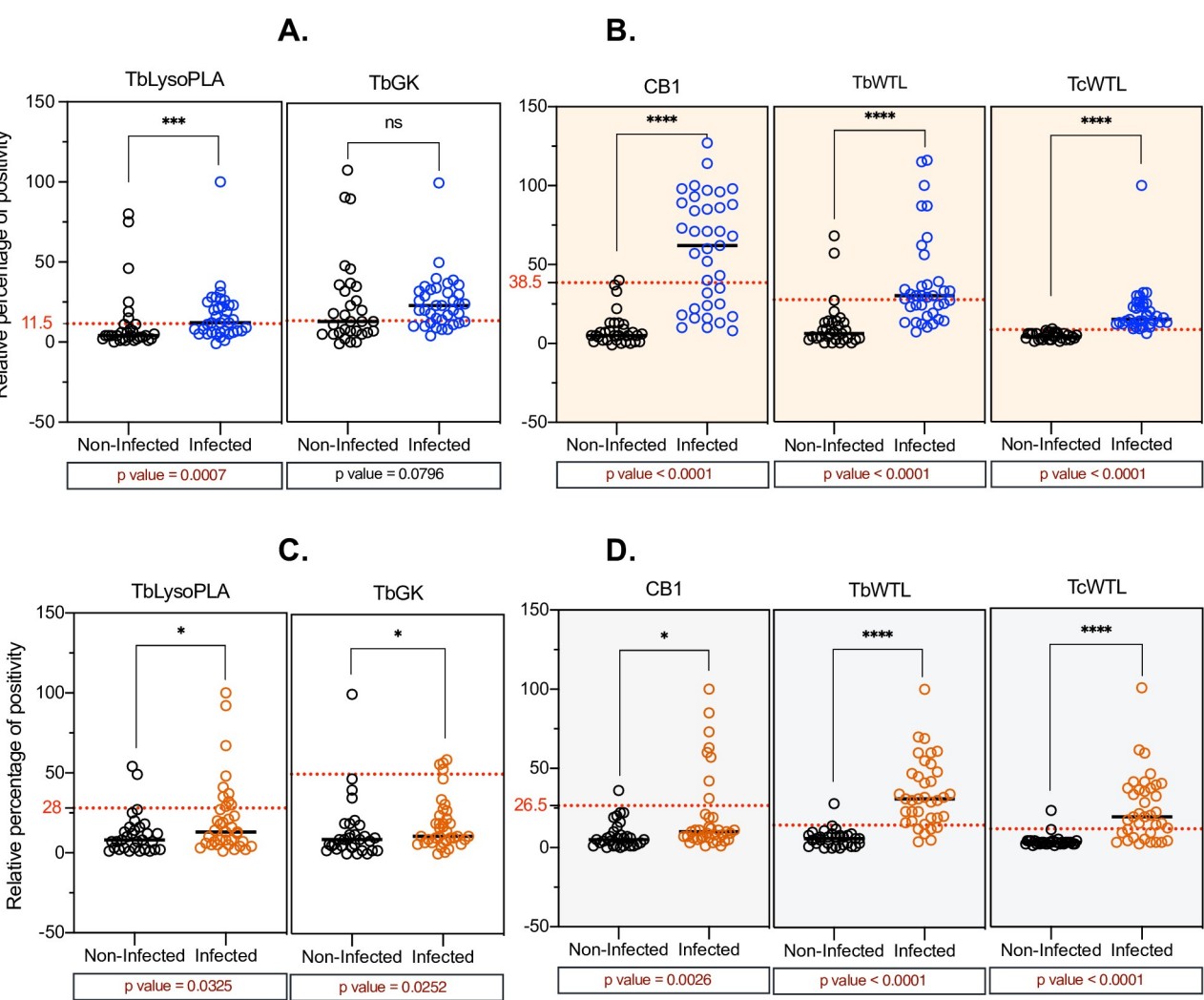

**Fig 3. Evaluation of the diagnostic potential of TbLysoPLA and TbGK in indirect ELISA. (A-B)** bovine sera from T. congolense experimental infection. (C-D) Field sera from T. congolense, T. vivax and T.b. brucei naturally infected cattle. A total of 74 positive (infected) and 62 negative (non-infected) samples were tested. TcoCB1 antigen was used as recombinant antigen control and TbWTL, TcWTL (T.b. brucei and T. congolense total lysates respectively) were used as reference antigens (current OIE recommended antigens in AAT diagnosis). The optimal cut-off value for each antigen is shown by a red dashed line.

positive rate (Se) versus false positive rate (1-Sp) and making visualization of the AUC has been established for each test (Fig 4B–4D). The latter is the best parameter in ROC analysis to get the global diagnostic performance.

Based on the AUC values, protein mixtures displayed the most accurate diagnostic performance compared to individual protein (Fig 4B–4D, Tables 4 and 5). In fact, the AUC value varies from 0.5 (powerless diagnostic) to 1 which is the most powerful diagnostic performance value. In Tables 4 and 5, the proteins tested individually show an AUC value ranging from 0.62 (TbGK) to 0.74 (TbLysoPLA) for TcEIC sera and from 0.65 (TbLysoPLA) to 0.66 (TbGK) for FS sera. These values are all lower than proteins combinations AUC values (≥0.78) which are close to 1. In regard to Se and Sp, TbGK showed the lowest Sp (54.8%) but an acceptable Se (73%) for TcEIC sera. TbLysoPLA showed a Se comparable to TbWTL reference antigen (~54%) and a satisfactory Sp of 80.6%. Both TbGK and TbLysoPLA displayed lower Se and Sp

**Table 4. Diagnostic performance parameters determined by ROC analysis of single antigen and mixture testings using sera of *T. congolense* experimentally infected cattle in ELISA.**

| | Cut-off | Se (%) | 95% CI | Sp (%) | 95% CI | LR | AUC | 95% CI |
|---|---|---|---|---|---|---|---|---|
| | | | Single antigen test | | | | | |
| TbGK | > 13.5 | 73.0 | 57.0–84.6% | 54.8 | 37.8–70.8% | 1.62 | 0.624 | 0.48–0.77 |
| TbLysoPLA | > 11.5 | 54.1 | 38.4–69.0% | 80.6 | 63.7–90.8% | 2.79 | 0.735 | 0.61–0.86 |
| | | | Combination test | | | | | |
| TbLysoPLA_TbGK | > 30.5 | 60.9 | 40.8–77.8% | 87.0 | 67.9–95.5% | 4.67 | 0.864 | 0.75–0.98 |
| TbLysoPLA_TbGK_C25 | > 24.5 | 43.5 | 25.6–63.2% | 91.3 | 73.2–98.5% | 5.00 | 0.821 | 0.70–0.94 |
| | | | Reference antigens | | | | | |
| CB1 | > 38.5 | 64.9 | 48.8–78.2% | 96.8 | 83.8–99.8% | 20.1 | 0.946 | 0.90–0.99 |
| TcWTL | > 8.50 | 97.3 | 86.2–99.9% | 96.8 | 83.8–99.8% | 30.2 | 0.991 | 0.97–1.00 |
| TbWTL | > 27.5 | 56.8 | 40.9–71.3% | 93.5 | 79.3–98.9% | 8.80 | 0.889 | 0.81–0.97 |

Cut-off: Threshold value dividing the test result into positive and negative

Se %: percent sensitivity and its 95% confidence interval (95% CI)

Sp %: percent specificity and its 95% confidence interval (95% CI)

AUC: Area under the curve and its 95% confidence interval (CI)

LR: Likelihood Ratio

values compared to the reference antigen TcWTL Se (97%) and Sp (96%) with TcEIC sera (Table 4 and S1E Fig). As hypothesized, the protein combinations Sp were higher at 87% for TbLysoPLA_TbGK (a) and 91% for TbLysoPLA_TbGK_C25 (b) compared to individual proteins. However, the Se was moderate for a (~61%) and lower for b (43%) (Table 4).

With field sera (FS), both TbLysoPLA and TbGK showed very high Sp values at 96.8 and 93.5% respectively, close or identical to the reference antigen Sp (96.8%) (Table 5 and S1F Fig), but Se were very poor (<28%). Conversely, protein combinations displayed a significant increase in Se (~61%) compared to individual proteins, while Sp for both remained satisfactory, 87% for a, ~96% for b (Table 5). In addition, the statistical analysis comparing the RPP

**Table 5. Diagnostic performance parameters determined by ROC analysis of single antigen and mixture testings with field sera (sera from *T.b. brucei*, *T. congolense* and *T. vivax* infected cattle) in ELISA.**

| | Cut-off | Se (%) | 95% CI | Sp (%) | 95% CI | LR | AUC | 95% CI |
|---|---|---|---|---|---|---|---|---|
| | | | Single antigen test | | | | | |
| TbGK | > 50.0 | 10.8 | 4.29–24.7% | 96.8 | 83.8–99.8% | 3.35 | 0.658 | 0.53–0.79 |
| TbLysoPLA | > 28.0 | 27.0 | 15.4–43.0% | 93.5 | 79.3–98.9% | 4.19 | 0.651 | 0.52–0.78 |
| | | | Combination test | | | | | |
| TbLysoPLA_TbGK | > 37.5 | 60.9 | 40.8–77.8% | 87.0 | 67.9–95.5% | 4.67 | 0.784 | 0.65–0.92 |
| TbLysoPLA_TbGK_C25 | > 17.5 | 60.9 | 40.8–77.8% | 95.7 | 79.0–99.8% | 14.0 | 0.857 | 0.75–0.97 |
| | | | Reference antigens | | | | | |
| CB1 | > 26.5 | 21.6 | 11.4–37.2% | 96.8 | 83.8–99.8% | 6.70 | 0.710 | 0.59–0.84 |
| TcWTL | > 11.5 | 67.6 | 51.5–80.4% | 96.8 | 83.8–99.8% | 20.9 | 0.900 | 0.83–0.97 |
| TbWTL | > 14.5 | 83.8 | 68.9–92.3% | 96.8 | 83.8–99.8% | 26.0 | 0.949 | 0.90–1.00 |

Cut-off: Threshold value dividing the test result into positive and negative

Se %: percent sensitivity and its 95% confidence interval (95% CI)

Sp %: percent specificity and its 95% confidence interval (95% CI)

AUC: Area under the curve and its 95% confidence interval (CI)

LR: Likelihood Ratio

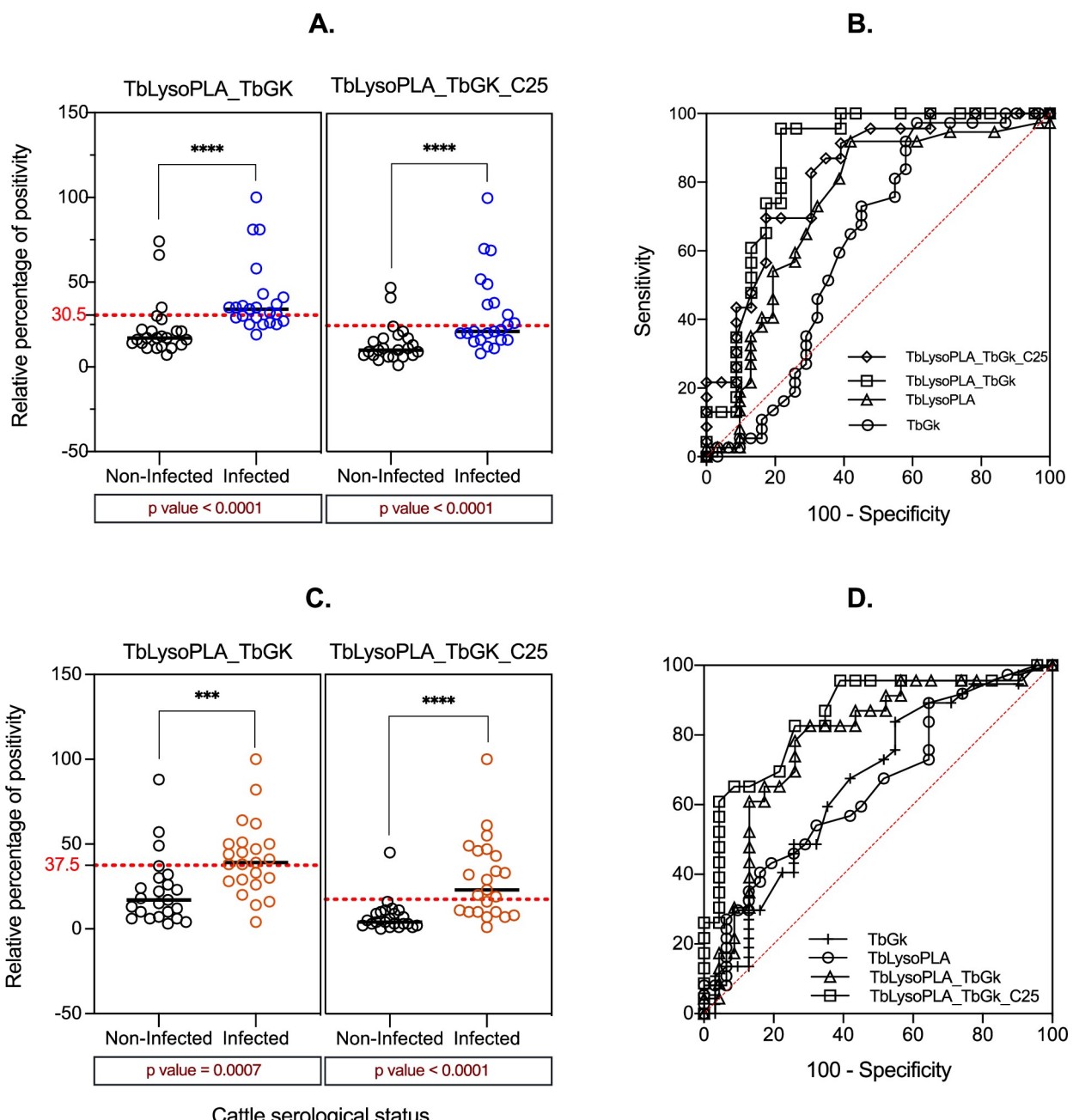

**Fig 4. Protein equimolar mixtures as antigen in Ab-ELISA and ROC curve comparison between individual proteins and mixtures.** (A.) bovine sera from T. congolense experimental infection (37 positive and 31 negative sera). (B.) ROC curve of antigens tested with experimental sera: TbLysoPLA (triangle), TbGK (dot), TbLysoPLA_TbGK (square) and TbLysoPLA_TbGK_C25 (rhombus). (C.) field sera from T.b. brucei, T. congolense and T. vivax mixed infection (37 positives and 31 negatives sera). (D.) ROC curve of antigens tested with field sera (same legends as in B). The optimal cut-off value (in A and C) for each antigen is shown in red dashed line. The global diagnostic performance was defined by AUC value.

medians of positive and negative samples on mixtures showed a highly significant difference between the two groups, contrasting with individual protein where the difference is weak (FS on both proteins) or not significant (TcEIC sera on TbGK) (Table 3).

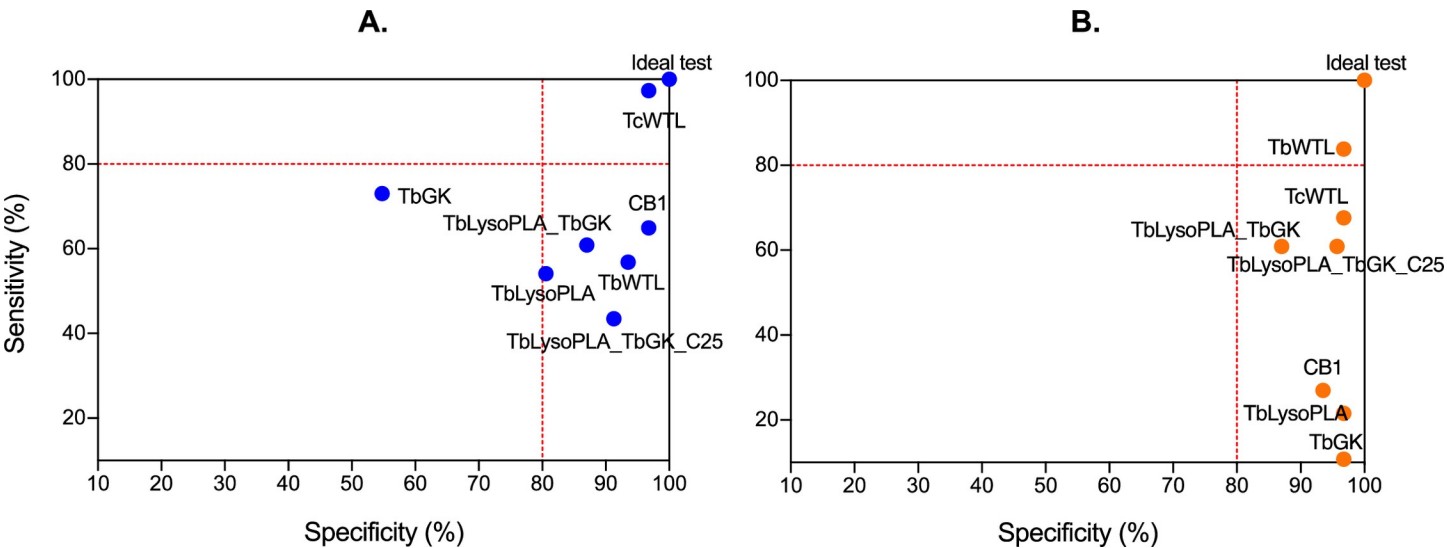

**Fig 5. Sensitivity and specificity relationship at optimal cut-off for each test and comparison with controls. (A.)** Bovine sera from experimental T. congolense infection. **(B.)** Field sera from T. b. brucei. T. congolense and T. vivax infected cattle.

In summary, recombinant protein mixtures used as antigens are much better than each protein tested individually, although they do not meet the TcWTL (for TcEIC sera) and TbWTL (for FS sera) control values regarding overall performance. Regarding the ROC curves, AUC value increased for the proteins combinations as shown in Fig 4B–4D. The Se/Sp pair for each protein and reference antigens are shown in Fig 5 to better highlight differences.

## Discussion

AAT is a severe constraint to livestock development in Sub-Saharan Africa. The problem is exacerbated by increasing resistance to trypanocidal drugs triggered by recurrent improper use of treatments (underdosing) due to insufficient support from veterinary services, and the widespread use of counterfeit products [6,27]. A vaccine would have been the most cost-effective method of control but vaccine has so far remained elusive for cattle [4,28]. A very recent study has shown the efficacy of an anti-*T. vivax* vaccine in mice [5] in a context of an experimental infection, but its potential application in cattle will require many additional developments. Surveillance is unsatisfactory and chemotherapy inefficient, for lack of the chief requirement, an adequate field diagnosis that follows ASSURED criteria defined by **A**ffordability, **S**ensitivity, **S**pecificity, **U**ser-friendliness, **R**apidity and robustness, be **E**quipment free, and **D**eliverable to end-users [29,30] or the newly REASSURED criteria defined by **R**eal-time connectivity, **E**ase of specimen collection, ASSURED [31]. While clinical signs are not pathognomonic [32], both microscopy and PCR do not meet ASSURED neither REASSURED criteria. Existing antibody detection tests based on whole trypanosome lysate are of limited use due to antigen standardization issues [33,34], though are open to improvement through recombinant technology. Several proteins have been evaluated for their diagnostic potential to replace total antigens [19,35–37], but few have made it to the RDT stage.

In the present study, we evaluated two new candidate proteins, the lysophospholipase (TbLysoPLA) and the glycerol kinase (TbGK), of *T.b. brucei* for AAT serodiagnosis. In addition, we provided the proof of concept of an innovative approach that consists of combining a set of recombinant proteins as antigens in a unique test. Using Ab-ELISA, we showed that the two proteins TbLysoPLA and TbGK had poor diagnostic performance when tested

individually, but drastically improved their sensitivity and specificity when combined with each other or with C25, particularly when using field sera.

TbLysoPLA and TbGK had never been tested for diagnostic purposes. However, we selected them to test their diagnostic potential based on their intrinsic properties, immunogenicity for TbLysoPLA, abundance for TbGK. *In silico*, both TbLysoPLA and TbGK proteins showed strong identities with their orthologs in trypanosomes (over 60% for TbLysoPLA and over 82% for TbGK). They displayed low homology with bovine (<35%) and *Piroplasma* (<15%) orthologous proteins. In addition, TbLysoPLA has seven predicted epitopes, including five shared with its ortholog in *T. congolense*. This strengthens the rationale for using it as an antigen in diagnosis with a low risk of cross-reaction with other parasite infections. The fact that TbGK shares a high identity with both cattle (58%) and *Piroplasma* (45–55%) suggests a low immunogenicity. It may not be the case however. TbGK presents on its sequence fourteen predicted epitopes, and it is not uncommon for certain conserved proteins to induce a strong antibody response, as was demonstrated for heat shock proteins, otherwise involved in protein maturation [19,38].

As a whole, the evaluation of a protein for AAT serodiagnosis usually consists in showing its specificity for a single species. This has been, among others, the case for TvGM6 of *T. vivax* [35], and TcoCB1 of *T. congolense* [21]. However, a test allowing a more comprehensive detection of the main pathogenic species such as *T. brucei s.l.*, *T. congolense* and *T. vivax* would be more useful in an epidemiological context. We have shown by western blot that both TbLysoPLA and TbGK orthologs of *T. congolense* are specifically recognized by specific antibodies directed against their orthologs of *T.b. brucei*. This makes possible the use of these proteins for the detection of *T. congolense* infections. Pertaining to their diagnostic potential, TbLysoPLA and TbGK taken individually showed poor performances, including a very low sensitivity with field sera (<28%), and a low specificity with experimental sera, 54% for TbGK, the latter quite unusual for a recombinant protein. Low sensitivity and mediocre specificity is quite typical of many proteins tested in serodiagnosis. For example, the rTES-30 protein showed only 30% of sensitivity for the detection of toxocariasis in cattle [26]. The HSP70 protein truncations tested for the serodiagnosis of *T. evansi* showed sensitivity values ranging between 41–61%, and specificity values between 65–91% [39], similar to our own results. Surprisingly, and despite these poor performances, the combination of TbLysoPLA and TbGK showed an improved diagnostic potential. We have shown that their equimolar mixture, combined or not with C25, was able to discriminate very clearly between infected and non-infected cattle groups, whether sampled during an experimental infection or in the field. Considering AUC as a global evaluation parameter, the mixtures showed AUC values of 0.78 to 0.86, higher than the proteins tested individually and closer to the values of the reference antigens testings which were up to ~1 for TcWTL applied on experimental sera. The individual protein testings showed a maximal AUC of 0.74 for TbLysoPLA and 0.66 for TbGK. This result is in agreement with other studies including the combination of RoTat1.2 and ISG75 in the serodiagnosis of *T. evansi* [40] and the combination of three *Leishmania infantum* proteins identified from cDNA in antibody detection ELISA [41].

Our study was limited by the small number of sera from *T.b. brucei* experimentally infected cattle available, and also by the absence of sera from *T. vivax* experimentally infected cattle. We used mainly sera from *T. congolense* infected animals to assess tests performances. The use of a *T.b. brucei* proteins to diagnose *T. congolense* or *T. vivax* infections could account in part for the low sensitivity observed, in particular with field sera on proteins used individually. Likewise, the hexa-histidine tag of TbGK had not been cleaved off the recombinant protein. The presence of this tag could affect the nature of the epitopes that TbGK possesses, and reduce its immunogenicity, as indeed tags can modify epitopes, especially conformational

epitopes [42]. Finally, we could not assess the cross-reactivity of TbLysoPLA and TbGK with sera from cattle infected with *Piroplasma* (*Babesia*, *Theileria*) for lack thereof.

In summary, the combined approach of bioinformatics and ELISA testing has allowed the identification of strong candidates for the serodiagnostics of trypanosomose. We showed for the first time in the context of AAT diagnosis especially in *T. congolense*, *T. vivax* and *T.b brucei* infection, the improved performance of a mixture as compared to the same proteins taken individually in an ELISA test. These mixtures of two or more proteins, while there is room for improvement, exhibit increased sensitivity and/or specificity than proteins tested individually. The still moderate sensitivity observed (~60%) can no doubt be improved by combining additional immunoreactive proteins or part ones in these mixtures. The use of synthetic peptides encompassing the most immunodominant epitopes would improve standardization as proposed by Magalhães *et al.* (2017) for leishmaniasis serodiagnosis. Finally, for the sake of ease and cost of production, and also of standardization, genetically-engineered chimeras composed of epitopes of several proteins can be envisaged.

## Supporting information

**S1 Fig.  (A-B).** C25 protein in single antigen Ab-ELISA test with experimentally *T. congolense-infected* bovine (TcEIC) sera (A) and field naturally *T. congolense*, *T. vivax* and *T.b brucei* infected bovine sera, field sera, FS (B). **(C-F).** ROC curve analysis: C25 (C-D), TcoCB1 and references antigens (TbWTL, TcWTL) testing with TcEIC (E) and FS sera (F).
(TIF)

**S2 Fig. TbLysoPLA, TbGK, C25, TbWTL and proteins mixtures as antigens in Ab-ELISA with experimentally *T.b. brucei* infected bovine (TbbEIC) sera.** The figures represent the RPP medians values with their 95% CI values for infected and non-infected bovine sera.
(TIF)

**S3 Fig. Multi-sequence alignment of TbLysoPLA with orthologs in African trypanosomes.** The IDs: Tb927.8.6390 (*Trypanosoma brucei*), TcIL3000_8_6240 (*T. congolense*), TvY486_0805980 (*T. vivax*).
(TIF)

**S4 Fig. Multi-sequence alignment of TbLysoPLA with orthologous proteins of *Trypanosoma congolense, Trypanosoma vivax, Leishmania, Piroplasma* parasites and *Bos Taurus*.** The IDs: Tb927.8.6390 (*Trypanosoma brucei*), TcIL3000_8_6240 (*T. congolense*), TvY486_0805980 (*T. vivax*), LINF_240024400 (*Leishmania infantum*), BBOV_IV005690 (*Babesia bovis*), TP02_0354 (*Theileria parva*), Q17QL8 (*Bos taurus*). The predicted epitopes in Table 2 for TbLysoPLA were annotated (by epitope number and position) on the protein sequence aligned with orthologs.
(TIF)

**S5 Fig. Multi-sequence alignment of TbGK with orthologs in African trypanosomes.** The IDs: Tb427tmp.211.3540-t26_1 (*Trypanosoma brucei*), TcIL3000_0_55170 (*T. congolense*), TvY486_0906060 (*T. vivax*).
(TIF)

**S6 Fig. Multi-sequence alignment of TbGK with orthologous proteins of *Trypanosoma congolense, Trypanosoma vivax, Leishmania, Piroplasma* parasites and *Bos Taurus*.** The IDs: Tb427tmp.211.3540-t26_1 (*Trypanosoma brucei*), TcIL3000_0_55170 (*T. congolense*), TvY486_0906060 (*T. vivax*), A4IBJ9 (*Leishmania infantum*), BBOV_I004170 (*Babesia bovis*), TP02_0725 (*Theileria parva*), ENSBTAP00000043922.3, ENSBTAP00000030898.4 (*Bos*

*taurus*). The predicted epitopes in Table 2 for TbGK were annotated (by epitope number and position) on the protein sequence aligned with orthologs.
(TIF)

## Acknowledgments

We are grateful to all technical stuff of CIRDES especially to S. Sylla, H. Sakande, M. Bamba and L. Millogo who facilitated the search and selection of samples among the numerous serum banks and their technical support.

## Author Contributions

**Conceptualization:** Magamba Tounkara, Alain Boulangé, Sophie Thévenon, David Berthier, Loïc Rivière.

**Data curation:** Magamba Tounkara.

**Formal analysis:** Magamba Tounkara.

**Funding acquisition:** Frédéric Bringaud, David Berthier, Loïc Rivière.

**Investigation:** Magamba Tounkara, Magali Thonnus.

**Methodology:** Magamba Tounkara.

**Supervision:** Alain Boulangé, Adrien Marie Gaston Bélem, Zakaria Bengaly, Sophie Thévenon, David Berthier, Loïc Rivière.

**Validation:** Alain Boulangé, Sophie Thévenon, David Berthier, Loïc Rivière.

**Visualization:** Magamba Tounkara.

**Writing – original draft:** Magamba Tounkara.

**Writing – review & editing:** Alain Boulangé, Frédéric Bringaud, Sophie Thévenon, David Berthier, Loïc Rivière.

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
