## [Decision Letter · Decision Letter 0]

16 Aug 2021

Dear Dr Riviere,

Thank you very much for submitting your manuscript "Novel protein candidates for serodiagnosis of African animal trypanosomosis: Evaluation of the diagnostic potential of lysophospholipase and glycerol kinase from Trypanosoma brucei" for consideration at PLOS Neglected Tropical Diseases. As with all papers reviewed by the journal, your manuscript was reviewed by members of the editorial board and by several independent reviewers. In light of the reviews (below this email), we would like to invite the resubmission of a revised version that takes into account the reviewers' comments. 

We cannot make any decision about publication until we have seen the revised manuscript and your response to the reviewers' comments. Your revised manuscript is also likely to be sent to reviewers for further evaluation.

Sincerely,

Alvaro Acosta-Serrano

Deputy Editor

Reviewer's Responses to Questions

**Key Review Criteria Required for Acceptance?**

**Methods**

-Are the objectives of the study clearly articulated with a clear testable hypothesis stated?

-Is the study design appropriate to address the stated objectives?

-Is the population clearly described and appropriate for the hypothesis being tested?

-Is the sample size sufficient to ensure adequate power to address the hypothesis being tested?

-Were correct statistical analysis used to support conclusions?

-Are there concerns about ethical or regulatory requirements being met?

Reviewer #1: Methods appear to be ok - some small sample sizes and were unavoidable.

Reviewer #2: This manuscript by Tounkara et al describes two novel protein candidates for the serodiagnosis of AAT, a devastating disease affecting sub-Saharan Africa. The objectives of the study are clearly stated, the manuscript is well-written, and the study follows a logical progression.

Sample sizes and statistics are fine in all cases except for the TbbEIC sample group. However, this limitation is openly discussed by the authors, and combined with the larger sample groups, the inclusion of TbbEIC sample analysis is still warranted. One key limitation is the lack of T. vivax samples in order to test effectiveness of the serodiagnostics assay with this species. However, as above, this limitation is highlighted by the authors.

I have no concerns about ethical or regulatory requirements.

Reviewer #3: Please see summary below

**Results**

-Does the analysis presented match the analysis plan?

-Are the results clearly and completely presented?

-Are the figures (Tables, Images) of sufficient quality for clarity?

Reviewer #1: Generally. Please see my recommendations

Reviewer #2: Overall, the analysis presented matches the analysis plan, and results are clearly presented. I have only a few minor comments:

The rationale for choosing the two proteins as diagnostic candidates is currently outlined in the first two paragraphs of the results section. I would consider moving these to the end of the introduction as there are no real "results" to present regarding the identification of these two protein candidates (for example, TbLysoPLA is described as a secreted protein that induces production of specific antibodies, but these data are from a different study). This would make the protein sequence analysis the first paragraph of the Results section.

To give a more detailed overview of the two protein candidates, it may help to include protein sequence alignments of T. brucei, T. congolense and T. vivax for LysoPLA and GK (supplementary figures). These alignments can also be linked to the antigenic peptide sequences outlined in table 2, which would help the readers to see how similar these regions are between the three trypanosome species (in fact, it would greatly support the statement made on line 269, that these regions are conserved in all three species).

This reviewer is not very familiar with the BepiPred-2.0 tool. In table 2, what is the rationale for the different threshold parameters (0.52 and 0.45 for TbLysoPLA and TbGK, respectively), and should these not be the same?

For clarity, would the authors be able to include Gene IDs for the TbLysoPLA and TbGK genes in the methods (currently TbLysoPLA only) and/or results sections (e.g. Table 2). It would also be beneficial to the reader to include gene IDs for the orthologs in T. congolense and T. vivax.

Do the authors have access to any other parasitaemia data for the samples used in this study, especially for the experimental samples? This would give an idea of the accuracy of the serodiagnostics analysis in comparison to conventional diagnostics such as microscopy.

Figure 1B - Whilst RNAi of TbGK clearly leads to a reduction in protein (~55 kDa), the PFR control is also significantly reduced. Can the authors explain this?

Line 420 - I believe PLOS NTDs does not accept references to "Data not shown" - therefore the data regarding shared epitopes of TbLysoPLA between T. brucei and T. congolense will need to be included, if available (perhaps in supplementary materials?).

Reviewer #3: Please see summary below

**Conclusions**

-Are the conclusions supported by the data presented?

-Are the limitations of analysis clearly described?

-Do the authors discuss how these data can be helpful to advance our understanding of the topic under study?

-Is public health relevance addressed?

Reviewer #1: Yes.

Reviewer #2: One aspect that is not discussed is whether these serodiagnostic candidates are also effective in diagnosing all three subgroups of T. congolense (Savannah, Forest and Kilifi), could the authors comment on this (I assume, perhaps wrongly, that T. congolense IL3000 was used in the experimental samples)?

Can the authors comment whether there is cross-reactivity with Trypanosoma theileri infections? This species is known to infect a range of bovine hosts, but is not always clinically relevant (PMID: 28903536).

From this study, it appears that the use of at least two proteins is a significant improvement on diagnostics, compared to using an individual protein. Is there an optimum number of proteins to include in a panel for serodiagnosis (i.e. if a third protein was included, would this improve the diagnostic capability even more)?

The authors provide a good discussion of the importance of novel serodiagnostics candidates, but I was wondering whether they can discuss any other protein candidates that may be suitable in addition to the two outlined in this study (for example, by taking into account previous secretome studies of T.b. brucei)?

Public health relevance is nicely addressed in the discussion section - especially the point that a comprehensive test encompassing the three main pathogenic trypanosome species is highly desirable.

Reviewer #3: Please see summary below

**Editorial and Data Presentation Modifications?**

Reviewer #1: (No Response)

Reviewer #2: There are several minor grammatical and/or spelling errors to address, as well as some missing references in the introduction:

Line 60 – A very recent study found that vaccination against T. vivax is possible using an invariant surface protein antigen (PMID: 34040257). This should be included in the introduction. 

Line 66 – Parasite-derived RNA can be detected during in vivo infections, as well as DNA and proteins (PMID: 30779758).

Line 68 – ….comparatively easy to perform but lacks sensitivity (missing the ‘s’ after lack).

Line 84 – Protein mixtures.. (don’t need the ‘s’ at the end of proteins).

Line 99 - Where applicable, it would be useful for the reader to know which strains were used for each species (e.g. T. congolense I assume is IL3000 and T. vivax is Y486?). Same goes for Table 1, for trypanosome strains.

Line 107 – There is a space after TcoCBP1 before the closing bracket that is not needed (also on line 337).

Line 146 – “As regards the TbGK protein, it was purified…”, consider revising (e.g. “With regards to the TbGK protein”)

Line 406 - "two novel proteins" - consider revising because the proteins themselves aren't novel, but as candidates for serodiagnosis they are novel.

Line 469 - "are way more sensitive" -> "exhibit increased sensitivity"

Reviewer #3: Please see summary below

**Summary and General Comments**

Reviewer #1: Review of Tounkara et al., “Novel protein candidates for serodiagnosis of African animal trypanosomosis: Evaluation of the diagnostic potential of lysophospholipase and glycerol kinase from Trypanosoma brucei”

In this manuscript, Tounkara and colleagues describe the assessment of two proteins from Trypanosoma brucei as candidate components of a serology-based diagnostic approach for use in animal disease surveillance and treatment campaigns. The two proteins, a lysophospholipase A and glycerol kinase, are immunogenic and generally abundant, suggesting that they would be reasonable candidates for such a tool. While the individual proteins alone performed poorly, while the two combined with a third previously described antigen were more promising. While none of the combinations reached the level of sensitivity of total antigen samples, the work does provide evidence that the right cocktail of recombinant antigens could be identified to yield a standardized and sensitive assay for use in animals. 

A. Major Concerns:

1. Line 72 – The sentence starting on line 71, “However, there is as yet no…, that would allow detecting active infections” is a little misleading. Serology-based tests often cannot distinguish between current and previous (particularly, previously treated) infections. This is antigen/antibody dependent but may limit he utility of a serological in treatment campaigns involving domesticated animals that may have multiple infections in a lifetime. Some comment should be included about this potential limitation.

2. Line 237 – The function of GK in trypanosomes is not entirely resolved. It is essential in BSF parasites grown in glycerol-rich media, but that may not be its only function given its abundance. To suggest it is a glycolytic enzyme is not exactly accurate – it is involved in anaerobic glucose metabolism but is not a classical component of glycolysis. 

3. Lines 264-269. It is unclear if the predicted epitopes are unique to the trypanosome proteins – some comment on this would enhance the value of this analysis. 

4. What do the tops of the blots in Fig 1B look like. They seem to be cropped differently than the other blots and gels. 

5. Fig. 4 – Some comment should be added to the text to address the difference in performance of the three-protein cocktail when the experimentally infected vs. field sera are analyzed

B. Minor Comments:

1. Abstract, line 35 – insert “other” before trypanosomes 

2. Line 59-60 – change to “to regularly change their glycoprotein coat, conventional vaccines have not yet been developed…”

3. Line 285 – (needs two commas) “…hexa-histidine tag, respectively, were…” 

4. Line 502 – should be “Fig. 5” not “Fig. 4”

5. Figs. 3 and 4 – “pourcentage” should be changed to percentage (or %) in the Y axis labels.

Reviewer #2: See above for my overall comments. This is an interesting study providing new candidates for the development of serodiagnostics to identify active AAT infections, which are sorely needed for animal trypanosomiasis. I feel a few minor details are required, but certainly no extra experiments.

Reviewer #3: This study by Tounkara et al. evaluates the potential of two recombinantly expressed T. brucei proteins, individually and in combination with the C25 antigen, for the serodiagnosis of African animal trypanosomosis (AAT). The authors show that, overall, in combination these antigens show improved performance in ELISA assays.

The data presented here are of a very high technical standard and will be useful for researchers in the area of AAT diagnostics. Some issues with data presentation and interpretation need to be addressed before publication, however.

1) When comparing conservation between orthologs, it is usually unclear if the authors are talking about percent identity or similarity (e.g. abstract, lines 257ff, lines 417ff). Please clarify this throughout the manuscript.

2) The authors may wish to qualify their comments on the potential of developing a vaccine against AAT (PMID 34040257)

3) Line 82: please provide references for C-25

4) Line 83: “...initially selected from the literature for their immunogenicity” - please provide references

5) Line 110: please provide the systematic TriTrypDB ID for BiP/C-25

6) Line 123: please provide the systematic TriTrypDB ID for TbGK

7) Lines 129ff: please clarify which procedures apply to TbLysoPLA vs. TbGK. As written, the text suggests in both cases lysates were loaded onto a glutathione sepharose column.

8) Line 181ff: Origin of the sera. According to the Methods, for the field samples it was known which trypanosome species an animal had been infected with. However, throughout the study this information is not being considered, although this might be very useful in assessing cross-reactivity. Please reconsider this decision, or discuss briefly why this information was not being taken into account. Perhaps sample sizes would have been too small to give statistically robust data?

9) Line 229: “Phospholipases are among the enzymes...” - please provide references.

10) Line 231: reference 15 seems wrong here

11) Table 1: the first column should read (row 1) TbLysoPLA, (row 2) TcLysoPLA, (row 3) TvLysoPLA, (row 4) TbGK, (row 5) TcGK, row 6 (TvGK). Also, the much lower conservation (0.2%) between TvLysoPLA and the Theileria ortholog compared to TcLysoPLA seems odd. Is this a typo? 

12) Line 266, immunogenicity scores: for those readers not familiar with these predictions, please provide context for these scores. Also, what do these findings imply or suggest with respect to the goal of the project?

13) Line 264ff / Table 2: it would be very useful to provide detailed information on the conservation of the predicted epitopes between the trypanosomatids and between T. brucei and Bos bovis

14) Lines 270/275: “Specific anti-TbLysoPLA...” - assessing specificity is the aim of this experiment

15) Lines 287 / 290: should that read “5 L” instead of “5 mL”?

16) Line 288 / Fig. 2A: if I understand correctly, TbLysoPLA was released from the column by cleavage with thrombin. Why is that protein (presumably ~36 kDa) not visible in the eluate? 

17) Line 295 (and elsewhere): according to the Methods section, 

18) Line 308 (similar statements are being made elsewhere, e.g. lines 312, 341, 444): “...demonstrating that TbLysoPLA discriminates between infected and non-infected cattle”. As written, this is misleading. The statement is true for discriminating between groups of infected vs. non-infected cattle, but for individual animals (as indicated by poor specificity and sensitivity) it is not. Please revise these statements to avoid ambiguity.

19) Line 336: please give systematic TriTrypDB ID for TcoCB1.

20) Line 377: “...TbLysoPLA and TbGK showed very high Sp values at 96.8% and 93.5%, respectively”. These values seem inconsistent with the data presented in Fig. 3C. If this was true, it should be possible to find a cut-off above which more than 90% of samples belong to the ‘infected’ category. The figure does not seem to support this.

21) Line 387: “...AUC value increased for the proteins (sic) combinations as shown in Fig 4B-4D”. This is not the case for the +C25 samples in 4B (Table 4), which have a lower AUC value compared to the -C25 sample.

22) Line 412: “...to levels comparable to reference tests”. This is an overstatement. As acknowledged elsewhere, whole cells lysates consistently performed better.

23) Line 446: “...showed AUC values of 0.78 to 0.86 higher than...” should read “...showed AUC values of 0.78 to 0.86, higher than...”, otherwise this is misleading.

24) Line 453ff: this is a good paragraph on limitations of the study. Can you please comment on potential of cross-reactivity with Leishmania, considering the degree of protein conservation? Presumably this is not a big concern in cattle?

25) Line 502. Fig. 5 legend mislabelled as Fig. 4.

26) Fig S2: panels B-F are not referred to anywhere in the manuscript. Also, why are the data presented differently from other figures? Please show individual data points, as in Figs. 3 and 4.

PLOS authors have the option to publish the peer review history of their article (what does this mean?). If published, this will include your full peer review and any attached files.

Reviewer #1: No

Reviewer #2: No

Reviewer #3: No
---

## [Decision Letter · Decision Letter 1]

4 Nov 2021

Dear Dr Riviere,

Thank you very much for submitting your manuscript "Novel protein candidates for serodiagnosis of African animal trypanosomosis: Evaluation of the diagnostic potential of lysophospholipase and glycerol kinase from Trypanosoma brucei" for consideration at PLOS Neglected Tropical Diseases. 

As with all papers reviewed by the journal, your manuscript was reviewed by members of the editorial board and by several independent reviewers. The reviewers appreciated the attention to an important topic. Based on the reviews, we are likely to accept this manuscript for publication, providing that you modify the manuscript according to the review recommendations. 

Before the manuscript is fully accepted, please address the minor concerns raised by reviewer #1 and reviewer #3. 

Sincerely,

Alvaro Acosta-Serrano

Deputy Editor

Reviewer's Responses to Questions

**Key Review Criteria Required for Acceptance?**

**Methods**

-Are the objectives of the study clearly articulated with a clear testable hypothesis stated?

-Is the study design appropriate to address the stated objectives?

-Is the population clearly described and appropriate for the hypothesis being tested?

-Is the sample size sufficient to ensure adequate power to address the hypothesis being tested?

-Were correct statistical analysis used to support conclusions?

-Are there concerns about ethical or regulatory requirements being met?

Reviewer #1: (No Response)

Reviewer #2: (No Response)

Reviewer #3: See summary below

**Results**

-Does the analysis presented match the analysis plan?

-Are the results clearly and completely presented?

-Are the figures (Tables, Images) of sufficient quality for clarity?

Reviewer #1: (No Response)

Reviewer #2: The authors have added several supplementary figures to the revised manuscript which satisfy the requests from myself and other reviewers to include further information on epitope sequence similarity. These additions have improved the manuscript.

Reviewer #3: See summary below

**Conclusions**

-Are the conclusions supported by the data presented?

-Are the limitations of analysis clearly described?

-Do the authors discuss how these data can be helpful to advance our understanding of the topic under study?

-Is public health relevance addressed?

Reviewer #1: (No Response)

Reviewer #2: (No Response)

Reviewer #3: See summary below

**Editorial and Data Presentation Modifications?**

Reviewer #1: (No Response)

Reviewer #2: (No Response)

Reviewer #3: See summary below

**Summary and General Comments**

Reviewer #1: Review of Resubmission of Tounkara et al., “Novel protein candidates for serodiagnosis of African animal trypanosomosis: Evaluation of the diagnostic potential of lysophospholipase and glycerol kinase from Trypanosoma brucei”

Overall, the authors have addressed the concerns I had with the manuscript. Just a few very minor clarifications/edits to recommend below:

1. The text found on lines 450-458 of the discussion is a bit confusing. “In silico, the two proteins showed …and low homology with bovine (<35%) and Piroplasma (<15%)” seems to conflict with line 458, “…TbGK shares a high identity with both cattle (58%) and Piroplamsa (45-55%)”. Please clarify this.

2. Line 112 – please change, “By maintaining...” to “When maintaining…”

Reviewer #2: The reviewers have clearly improved their manuscript, and taken my comments, as well as those of the other reviewers, into account. Thanks especially for their clarity in their answers. I feel this manuscript is now suitable for publication.

Reviewer #3: My comments have been addressed to my satisfaction, with two exceptions. I find the stated reason for not being able to provide the systematic ID for TcoCB1 (point 19) very odd, considering that the first author of one of the original studies is second author on this manuscript. And I still don't understand how the data shown in Fig 3C are consistent with the stated specificities (point 20). But considering that all the other comments have been addressed, I'll leave it to the Editor to decide is another round of revisions is necessary.

PLOS authors have the option to publish the peer review history of their article (what does this mean?). If published, this will include your full peer review and any attached files.

Reviewer #1: Yes: James Morris

Reviewer #2: No

Reviewer #3: No

Figure Files:

Data Requirements:

Reproducibility:

References

---

## [Editor Report · Decision Letter 2]

8 Nov 2021

Dear Dr Riviere,

We are pleased to inform you that your manuscript 'Novel protein candidates for serodiagnosis of African animal trypanosomosis: Evaluation of the diagnostic potential of lysophospholipase and glycerol kinase from Trypanosoma brucei' has been provisionally accepted for publication in PLOS Neglected Tropical Diseases.

Best regards,

Alvaro Acosta-Serrano

Deputy Editor

---

## [Editor Report · Acceptance letter]

9 Dec 2021

Dear Dr Rivière,

We are delighted to inform you that your manuscript, "Novel protein candidates for serodiagnosis of African animal trypanosomosis: Evaluation of the diagnostic potential of lysophospholipase and glycerol kinase from Trypanosoma brucei," has been formally accepted for publication in PLOS Neglected Tropical Diseases.

Best regards,

Shaden Kamhawi

co-Editor-in-Chief

Paul Brindley

co-Editor-in-Chief
